# An Assessment on the Efficiency of Clothing with UV Protection among the Spanish Navy School Students

**DOI:** 10.3390/ma15186227

**Published:** 2022-09-08

**Authors:** José Roberto Ribas, Sol García Rodríguez, Elena Arce Fariña, Andrés Suárez-García

**Affiliations:** 1Escola Politécnica, Universidade Federal do Rio de Janeiro, Rio de Janeiro 21941-909, Brazil; 2Research Group Solar and Wind Feasibility Technologies (SWIFT), Electromechanical Engineering Department, University of Burgos, 09006 Burgos, Spain; 3Department of Industrial Engineering, Ferrol Polytechnic University School, University of A Coruña, 15403 A Coruña, Spain; 4Defense University Center, Naval Academy, University of Vigo, Plaza de España 2, 36920 Marín, Spain

**Keywords:** Spanish navy, ultraviolet protection, uniforms, perceived efficiency, structural equation model

## Abstract

Concern about the harmful effects that ultraviolet (UV) rays have on the skin of people who are routinely exposed to solar radiation has driven the industry of skin protection creams, sunglasses and clothing. Spanish Navy personnel are subject to different levels of exposure depending on their rank and function. The objective of this research is to analyze the behavioral variables associated to the effects on the skin caused by UV rays, denoted by the combined effects of perceived susceptibility and perceived severity, on their decision to purchase and wear uniforms with UV protection. A confirmatory analysis using a structural equation modeling (SEM) was performed on a sample of 100 respondents. The model results revealed a strong mediating characteristic of the intention to use, variable associated with the exogenous variables. Attitude towards the use of clothing and social influence, as well as the exogenous variable clothing action planning, on the sun protective clothing use during tactical maneuvers. These relationships were significant with *p*-values close to zero. However, exogenous variables related to perceived susceptibility and perceived severity in exposure to sunlight did not represent a significant influence when mediated by self-efficacy in use. The results revealed the consequence of awareness about the importance of protecting oneself and the influence that usage habits can have on the military with respect to the decision to purchase uniforms with UV protection.

## 1. Introduction

People whose work involves a lot of exposure to the sun suffer to a greater extent from the consequences of UV radiation. Among these groups of people, the military stands out since, during maneuvers, for example, the military is exposed for all or practically all day and, although it is not always the case, they must train in it, so sooner or later they will have to deal with this problem. Their maneuvers can be performed in places where the solar intensity is high, such as the sun belt region between 35° S and 35° N, where troops may be subjected to high ultraviolet index levels. Moreover, the daily patrols in the area of operations with no more shade, in most cases, than that provided by the garments they wear, the ideal conditions are combined to suffer the consequences of the impact of the UV radiation. Studies for the United States Armed Forces have shown that those who have served in the military have a higher risk of skin cancer than those who have not [1]. Among the results, it should be noted that the members with the highest rate of skin cancer are the pilots of the “US Air Force”, both in melanoma and non-melanoma cancer. This is because, as has already been stated in the Introduction section, the higher the altitude, the greater the irradiance on our body. In fact, the level of UV radiation increases by around 15% for every 1000 m of altitude. Another unit clearly affected by this issue is the “10th Mountain Division” of light infantry, belonging to the US Army. This unit has a high rate of skin cancer due to the amount of time they spend training in the mountains at high altitudes. On the other hand, at the other extreme, the lowest rate is found among military personnel assigned to mechanized units, due to the lower exposure they suffer from being covered most of the time. These results are applicable to the Spanish Armed Forces since, being allies of the United States, they share most of the deployments and areas of operations [2,3].

In most units of the Spanish Navy, the activities carried out there are largely abroad, especially during periods of instruction and training [4,5]. Students undergo a rigorous training program, which exposes them not only to summer heat and sun but also to harsh winter conditions at high altitudes, where the sun is even more intense and UV radiation more concentrated. In addition, marine army corps are subjected to high levels of reflected solar radiation from the sea. Moreover, their activities can be performed in areas of the world close to the Equator, where the impact of UV radiation is much more dominant, such as regions of Africa or the Middle East.

Although the students wear the uniform camouflaged with a cap, hat or helmet as headwear, in the case of the Marines, or the fire-retardant task with a cap in the case of the General Corps and Quartermaster students, there are still areas exposed to radiation, such as the neck and face. The impact of UV radiation is cumulative on our skin [6], so the consequences of overexposure to radiation on our health, whether skin, eye or the immune system, can occur in certain students when over time, evidencing themselves in the performance of their duties as future officers of the Navy.

As for the seafarers, they are not exempt from this problem either. In fact, at sea, the reflection produced by UV rays means that being outdoors, on a rowing boat or on maneuvers, for example, can become more damaging than a patrol in the desert. In fact, the use of sunglasses at sea is more necessary, if possible, than on land. Despite this, similar to the military in land operations, the uniform gear corresponding to sailors also serves to protect themselves from the sun, since headwear should be worn outdoors, such as caps in most countries, cases and long-sleeved fire-retardant material [7].

The incidence of skin cancer is continuously increasing and has reached high proportions worldwide [8,9,10,11,12]. The term “skin cancer” includes a group of malignant neoplasms with very different characteristics (i.e., origin, risk factors and prognosis). There are the following two main groups: cutaneous melanoma and non-melanoma skin cancer (NMSC). NMSC includes squamous cell carcinoma and basal cell carcinoma. NMSC is the most frequently diagnosed group of malignant neoplasms in whites [13].

Clothing provides sun protection by absorbing, scattering and reflecting part of the UV radiation that reaches the surface of the fabric [14]. Fabrics differ in their capacity to reduce UV incidence because they differ in fiber composition and moisture content [15]. According to the Skin Cancer Foundation, the UV protection provided by clothing depends on [16]:-The porosity, weight and thickness: UV protection is higher the smaller the spaces in the yarns (weave) and the greater the weight and thickness of the fabric [17];-Color: darker colors offer higher UV protection;-Tension: UV protection decreases as the fabric stretches;-Humidity: UV protection decreases when the fabric is wet;-Washing: washing new garments increases UV protection, especially when they are made of natural fibers (e.g., cotton) [18].

The *Australia/New Zealand Standard for Protective Clothing* (AS/NZS 4399:1996) was the first published standard on methods for determining the ultraviolet protection factor (UPF) of clothing designed for this purpose [14]. This standard divides UPFs into the following three groups: those with ratings from 15 to 24, those with ratings from 25 to 39 and those with ratings between 40 and higher, which offer outstanding protection [19]. In addition to the Australian and New Zealand committees, the International Organization for Standardization (ISO) and other global organizations, such as the Commission on Illumination (CIE), have been working on developing UV-protective fabric standard papers. A new standard concerning the specifications for test procedures and labeling of sun-protective clothing (EN 13758) has been created by the European Committee for Standardization (CEN) [19]. The amount of ultraviolet light that is transmitted or blocked by textile materials designed to be used for UV protection can also be measured using the American Standard (AATCC TM 183) the GBT18830-2009 national standard of the People’s Republic of China, and the UV Standard 801 from the International Testing Association in Europe [20,21]. The UPF results by following these measurement standards do not differ significantly [20].

Ultimately, the problem does not lie in the deficiency of means to be offered by the Armed Forces, but rather in the employment that the military decides to give it. According to a study carried out by the American Academy of Dermatology in 2017 on previously deployed military personnel on a mission, less than 30% of the soldiers interviewed claimed to have used sunscreen regularly, and more than 70% had unprotected skin. The level of awareness must be high, and it must be highlighted among the members of the Armed Forces, since the consequences of the impact of UV radiation on the military are greater than on the rest of the ordinary citizens, and it has been demonstrated. According to another study, this time carried out by the National Cancer Institute of the United States in 2010, it was shown that the index of melanoma diagnosed in patients belonging to the military was higher than the index present in civilian citizens, especially in the age band between 55 and 59 years of age, further demonstrating that the damage caused by overexposure to UV radiation is cumulative, and its consequences do not appear until later stages [1].

Therefore, this problem must be considered in the employment of the military, since the simple fact of being assigned to an area closer to a forest, in the sea, the desert or higher in the mountains implies a factor of considerable risk that cannot be ignored or ignored, since the consequences, if protective measures are not taken, can be lethal.

The literature on the use of SEM to analyze problems with UV radiation in people regularly exposed to the external environment has addressed different objectives. A study was carried out with 148 male external workers with the objective of confirming that these professionals highly exposed to UV radiation could count on an efficient preventive diagnosis, obtained through interventions with UV photographs [22]. The investigation into the beliefs of 334 students at an American university about UV exposure and sunscreen use, as well as the links with skin cancer and protective measures in a cloudy sky [23]. Adoption of the tripartite theory of body image, reasoned action, health belief model and motivational protection theory to test various behavioral models. In this model, the authors confirmed that intention is a mediator of sociocultural influence and tanning behavior [24]. A model to measure the behavior of 273 parents in protecting their children from the carcinogenic effects of sun exposure. The variables used were intention, attitude, subjective norm, action plan, among others [25]. The use of Latent Class Analysis (LCA) to assess classified homogeneous groups of 90 respondents according to constitutional risk, tanning intention, and sun protection behavior (shade, clothing and sunscreen) [26]. The collection of a sample of 261 respondents to propose a model based on knowledge, motivation, health literacy and nutritional literacy to test whether knowledge about health and nutrition was associated with an attitude towards sun exposure [27].

The objective of this research is to analyze the behavioral variables associated with the effects on the skin caused by UV rays, denoted by the combined effects of exogenous variables, on their decision to purchase and wear uniforms with UV protection. The students’ perception of the attitude toward uniforms with UV protection should inspire manufacturers to make this option available, showing that the company’s efforts will be rewarded with the purchase decision made by students.

## 2. Materials and Methods

### 2.1. Structural Equations Model

In its most simplified form, SEM can be understood as the combination of path analysis with factor analysis [28,29] to enable the understanding of complex patterns of interrelationships. In path analysis, the concern is with the causal path of the observed variables [30,31]. In a complete structural equation model, the interest is in the causal path of the constructs (latent variables or factors). Although the acceptance of SEM is growing, the difficulties of its employment should not be underestimated [32]. Variables that are not directly measurable, which are supposedly the factors that cause a set of observed variables (measured or indicators) to vary, are usually called latent variables (or factors). As the latent variable is not measured, it is necessary to give it a metric, which is performed when the value 1 is assigned to one of the paths directed to one of the observed variables that it influences. With this restriction, the remaining paths can be estimated. When each latent variable has only one indicator, the SEM is restricted to path analysis. When each variable has multiple indicators but there are no direct effects connecting them, there is a factor analysis. However, when there are multiple indicators for each latent variable (or factor), as well as paths connecting them, complete structural models are built.

In SEM graphs, the linear regression model implicitly assumes zero measurement error. The error terms of the observed variables are explicitly modeled in SEM. Ignoring such errors can lead to serious distortions, especially when they are significant. Note that these terms should not be confused with residual error terms, also called disturbance errors, which reflect unexplained variance in latent endogenous variables due to unmeasured causes. Structural equation models also include possible direct and indirect effects between their variables [33]. Bearing in mind that a complete structural equation model is the combination of factor analysis and path analysis, it exhibits the following two intrinsically interrelated sections: (i) measurement; (ii) structural. The measurement section of the model corresponds to the factor analysis, and it describes the relationship between latent and observed variables. The measurement model is generally used as an independent (or null) model. In fact, the covariance in the latent variable covariance matrix of the independent model is considered null. The structural section of the model corresponds to the path analysis and represents the direct and indirect effects of the latent variables on each other. In the traditional path analysis, the aforementioned effects occur between observed variables; in SEM, path analysis is fundamentally between latent variables [34].

### 2.2. Variables Specification and Assessment

This study has focused on a select of students from a military school in Spain that was undergoing outdoor training at the time it was carried out. These respondents answered a questionnaire containing behaviorally observed variables related to the acquisition and use of military clothing with protection against UV rays. To avoid interpretation bias, an explanation was given on the meaning of UV protection from the sun in military clothing and aspects such as price variations, breathability, fit and lightness relative to unprotected clothing. The survey for measuring behavior towards the use of military sun protective clothing (Table 1) was an adaptation of the questionnaire developed by Van Osch et al. [27] to predict the parental use of sunscreen. The survey was distributed to students at the military naval school through Google Forms. Each of the questions was translated into Spanish from the work of Van Osch et al. [27], adapting sunscreen to sun protective clothing and the target audience to military and subordinates. The first version of the questionnaire was found to be difficult to comprehend and therefore a number of improvements were made to it such as better consolidation of the questions and provision of some examples to better explain the questionnaire.

### 2.3. Model

At this point, an explanation about the latent variables, unobserved exogenous variables and observed variables must be performed. In Figure 1, a high-level flow chart of the methodology is provided. Three latent and five observed variables were applied in this model. Attitude, social influence and action planning were the latent variables. Perceived susceptibility, perceived severity, self-efficacy, intention and sun protection clothing use the observed variables. This workflow model is based on the work of Van Osch et al. [27]. As indicated above, the questionnaire and the model of the relationships between input variables used in the research by Van Osch et al. [27] had to be translated into Spanish and adapted to the objective of the study as follows: to analyze the effectiveness of the use of UV protective clothing in students linked to the military. In other words, Figure 1 shows the hypothesis proposed in the model of dependencies between the variables used in this study.

### 2.4. Participants

The student sample consisted of 127 students from the Naval Military School of Marín (Spain), among whom 80 were from the General Corps, 39 from the Marine Corps and 8 from the Intendancy. Of the participants, 93.70% were male (119 students) and 6.30% were female (8 students). The age of the participants aged from 18 to 36 years (M = 22.52, SD = 2.90).

## 3. Results

Initially, all measurements underwent assumption tests characteristic of studies involving multivariate analysis [34]. The exogenous observed variables perceived susceptibility (SUS) and perceived severity (SEV), and the latent variables attitude (ATT), action planning (PLN) and social influence (SOC) revealed the absence of multicollinearity. The assumptions of normality and multivariate linearity resulting from the two-by-two combinations of metric variables were met. In order to preserve the model’s internal consistency, it is necessary to verify the distinctive robustness between the ATT, PLN and SOC constructs. Adopting the line of reasoning operationalized by Hagger and Chatzisarantis [35], the robustness between the three dimensions was evaluated.

The ATT, PLN and SOC latent variables showed reliability measured by Cronbach’s Alpha equal to 0.733, 0.866 and 0.558, respectively, all but one exceeding the minimum value of 0.70 according to Fornell and Larcker [36] and, therefore, meeting the reliability requirements. The chi-square of 307.8 divided by 116 degrees of freedom equal to 2.653 is satisfactory as it surpasses the more conservative significance level of 0.01. After verifying the inexistence of multicollinearity between the ATT, PLN and SOC, the estimate of the hypothetical theoretical relationships was carried out with the support of a structural equation model (SEM) adjusted on the latent variables as can be seen in Figure 2.

The concern when the theoretical model was initially conceived was about the apparent transposition that ATT could present with PLN and SOC, which, despite the care taken in the semantic conception of the questions, incorporated the risk of interpretation by the respondent. The rationale of the test states that, for instance, if ATT does not turn out to have distinct dimensions of PLN and SOC, causing a forced solution from two factors instead of three, should result in ATT collapsing all or part of the two other constructs’ direction. On the other hand, if SOC collapses to the factor where ATT is found or to the factor where PLN is found, we will have the latter two prove their mutual distinction and, therefore, the successful interpretation by the respondent. The obtained results with a two-factor analysis showed that the respondent discriminated between the two constructs since one of the factors incorporated the factorial loads of PLN and SOC, while the other kept the factorial loads of ATT isolated.

Then, the three dimensions underwent an exploratory factor analysis with the aim of validating the hypothesis of the non-occurrence of independence that would harm the isolated explanatory contributions of the covariance among the constructs. A varimax rotation with Kaiser normalization was adopted for the three-factor extraction. The factor loadings resulting from a cut applied to the solution for three factors, whose eigenvalues captured 60.3% of the extracted variance, are shown in Table 2, as well as the corresponding *t*-tests, all significant at the 5% bilateral significance level. Such results denoted the discriminant behavior in the interrelationship between dimensions. It can be seen that factor 1 refers to the observed variables 20 to 23, confirming the latent variable PLN. Likewise, factor 2 refers to the observed variables from 10 to 14, confirming the latent variable ATT. Factor 3 validated two of the three variables observed for the latent variable SOC.

## 4. Discussion

Regarding the causal relationship that the exogenous variables susceptibility (SUS) and severity (SEV) have over the mediating variable efficiency (EFF), the parameters were estimated as 0.271 with a *p*-value of 0.028 and 0.291 with a *p*-value of 0.084, respectively. This confirms hypotheses H_0_ and H_1_ at a 10% of significance level, that protecting oneself from the sun prevents the risk of developing skin cancer in the future. Furthermore, in the event of a positive diagnosis of skin cancer, wearing sun-protective clothing would be an effective way of protecting oneself from the sun.

The hypothesis that the mediating variable EFF, together with the latent variables ATT and SOC, may predict Intention (INT) resulted in the following parameters: 1.078, 0.365 and −0.398, respectively. The relationships between EFF→INT and ATT→INT were significant, having their *p*-values close to zero and confirming hypotheses H2 and H4, while SOC-INT was not significant. It means that the positive attitude towards wearing UV-protected clothing for someone to protect from the sun, combined with the perception that wearing clothing with sun protection would be effective, would positively influence the willingness to wear sun protection clothing to protect from the sun in military activities. Additionally, the social norms that recommend the use of protective clothing do not influence the intention of a military to use this type of clothing.

Finally, in evaluating the influence that the EFF, Action Planning (PLN) and INT variables have on the habit of wearing sun protection clothing (HAB), the parameters were estimated at 0.004, 0.485 and −0.261. The PLN→HAB and INT→HAB relationships were significant with *p*-values close to zero, confirming hypotheses H7 and H6; however, the mediator variable EFF did not show a significant relationship with HAB. This means that when a person plans to wear an outfit with sun protection, one is likely to convert this idealization into a habit of use. On the other hand, the predisposition to wear clothing with sun protection is inversely related to the habit of wearing it, which probably means that the mediator variable INT is treated as a motivation that has yet to materialize, while the endogenous variable HAB was perceived as an action incorporated into a habit of use, no longer requiring reinforcement of motivation. The failure of the EFF variable as a mediator confirms the possibility that the military personnel interviewed do not perceive the existence of a relationship between the belief that clothes with sun protection are effective and their habit of use, probably because the frequent use of an outfit is an already incorporated action that does not require cognitive reinforcement.

## 5. Conclusions

Modeling with the use of latent variables allowed non-measurable variables, such as beliefs, motivations and attitudes, to compose an explicit structure regarding the use of uniforms with UV protection. The use of measurable scales to reflectively compose latent variables is obviously subject to design and measurement errors; however, mitigated by the statistical rigor in the internal consistency test. The causal relationships involving exogenous variables and latent variables proved to be very useful in explaining the motivational structure and in identifying relevant elements for the adoption of managerial actions that could stimulate interest in uniforms with UV protection.

The results showed that, in general, the responses of the Spanish Marine Corps student sample showed positive manifestations regarding the attitude and decision to buy and use uniforms with UV protection. The results found in the structural equation model partly answer the problem of this research; that is, that the interviewee now has a favorable intention toward the clothing after realizing through the social influence that it provides safety as a positive and uses it in the purchase and use decision. This means that the protective benefit against the harmful effects of the sun’s rays is perceived by the interviewee based on their intention to use it and the influence exerted by people in their social life, which are captured in the model by the specified latent variables.

As a practical implication, it is possible to observe that uniforms with protection against UV radiation translate into a feeling of security and, therefore, constitute an attribute that suggests investment on the part of the student, resulting from the utilitarian awareness of clothing and the intrinsic relationship with the feeling of worth. With this, it is noted that the attitude, in addition to encouraging preventive behavior, also contributes to the confirmation of the social influence on the favorable intention towards clothing. The importance of the reputation that UV protection garments have as a focal point for purchase and use intentions by students is highlighted. However, there are no great efforts by suppliers of military uniforms to offer products with this type of protection. Therefore, drawing a parallel with the theory, students’ perception of the attitude toward uniforms with UV protection should encourage manufacturers to make this option available, showing that the company’s efforts will be rewarded with the purchase decision made by students. This situation demonstrates the need for further studies to investigate and deepen the evaluation of the loyalty process.

As academic contributions, this research provided support for the literature on the connection that occurs between knowledge about the harmful effects of UV radiation, the intention to use UV protective clothing and the consequent purchase intention and decision by Spanish Marine Corps students. Studies on the variables that influence the demand for uniforms with UV protection are extremely relevant when it is concluded that the effect of UV radiation affects an extremely valuable intangible asset, namely, the health of students.

## Figures and Tables

**Figure 1 materials-15-06227-f001:**
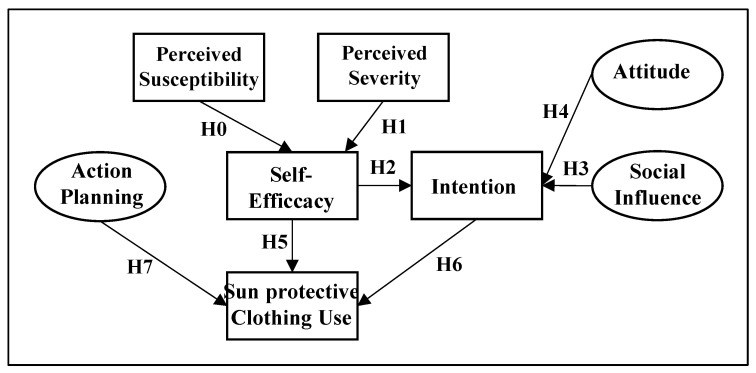
Proposed model for the UV textile usage.

**Figure 2 materials-15-06227-f002:**
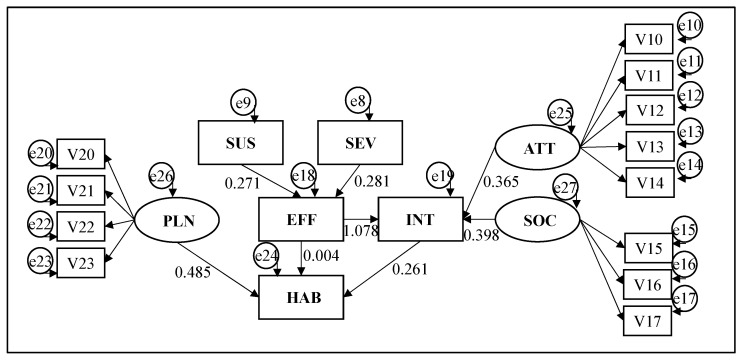
SEM specification for the UV textile usage.

**Table 1 materials-15-06227-t001:** Description of the measurement of cognitive concepts and military sun protective clothing use.

Concept	Number of Items	Item Description. Questions
Perceived susceptibility	1	If a military member does not protect himself or herself from the sun, the risk of suffering from skin cancer in the future is from very low (1) to very high (5)
Perceived severity	1	2. If a military member would diagnosed skin cancer in the future, I would find this from not serious (1) to very serious (5)
Attitude	5 (α = 0.75)	3. Wearing sun protective clothing during maneuvers is a comfortable way to protect the military from the sun. (1 = not pleasant to 4 = very pleasant) 4. Wearing sun protective clothing for a military member health is: (1 = not important to 4 = very important) 5. Wearing sun protective clothing is annoying during daily life in the military. (1 = very annoying to 4 = not annoying) 6. Wearing sun protective clothing is inconvenient, because I tend to forget it. (1 = very inconvenient to 5 = not inconvenient) 7. Wearing sun protective clothing is unnecessary. (1 = very unnecessary to 4 = not unnecessary)
Social influence	3 (α = 0.73)	8. *Social Modeling*: How often does the military wear sun protective clothing? (1 = hardly ever to 4 = always) 9. *Social Norms*: How important do important military in your environment find it to wear sun protective clothing to protect themselves from the sun? (1 = definitely not important to 4 = definitely important) 10. *Social support*: How important do important military in your environment find it to wear sun protective clothing to protect you from the sun? (1 = definitely not important to 4 = definitely important)
Self-efficacy	1	11. Do you think you will be able to adequately wear sun protective clothing to protect you from the sun? (1 = definitely not to 7 = definitely yes)
Intention	1	12. Do you intend to adequately use sun protective clothing to protect you from the sun? (1 = definitely not to 7 = definitely yes)
Action planning	4 (α = 0.86)	13. Do you plan to buy sunscreen sun protective clothing? 14. Do you plan to bring sun protective clothing with you during military exercises on a sunny day? 15. Do you plan to keep track of time in order to wear sun protective clothing depending on the weather? 16. Do you plan to ask other military to remember you to wear sun protective clothing? (all answers: 1 = definitely not to 7 = definitely yes)
Sun protective clothing use	1	17. How often did you protect yourself from the sun by wearing sun protective clothing when you were outside on a sunny day during past military exercises? (1 = never to 5 = always)

**Table 2 materials-15-06227-t002:** Factor loads after varimax rotation with Kaiser normalization.

Variable	Factor 1	Factor 2	Factor 3
V10	0.127	0.749	0.113
V11	0.189	0.650	0.091
V12	0.179	0.726	−0.153
V13	0.013	0.671	−0.012
V14	0.152	0.645	0.187
V15	0.224	−0.007	0.828
V16	0.482	0.140	0.179
V17	0.113	0.121	0.861
V20	0.762	0.244	0.237
V21	0.855	0.090	0.053
V22	0.871	0.046	−0.015
V23	0.783	0.219	0.146

## Data Availability

The data that support the findings of this study are available from the corresponding author, upon reasonable request.

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
