# Peer review of "An Assessment on the Efficiency of Clothing with UV Protection among the Spanish Navy School Students"

_materials, 2022, doi:10.3390/ma15186227_

Round 1

Reviewer 1 Report

Thank you for submitting your manuscript to Materials Journal.

Please find below some comments for your consideration:

The abbreviation IUV should probably be UVI = Ultraviolet Index.

Page 2 line 38: "Without going any further, in countries very exposed to the Sun such as Mali, where the Marine Corps currently has deployed troops, or Iraq (Spanish Ministry of Defense, 2020), where there are also troops, UV radiation can be very dangerous, reaching very high IUV levels." Kindly reconsider this sentence: First, there is no need to pin point places where troops are deployed. One can just say in places where the solar intensity is high, and especially within the sun belt region (between Latitude 35° S and 35° N), troops may be subjected to high and prolonged exposure to solar UV intensity.

There is a lot of unnecessary detail, which may be subject to confidentiality. Did you get permission from the academy to publish such details? One week training, full day in the sun, etc....? I suggest that you review the whole paragraph: "In most units of the Spanish Navy, the activities carried out there are largely abroad, 58 especially during periods of instruction and training [4,5]. Even though Galicia is one of 59 the regions of Spain that receives the least sunstroke per year, this work has aimed to focus 60 on the students of the Marín Naval Military School (ENM). In the case of Marine Infantry 61 students, they spend at least one full day in the field each week of the academic year, in 62 different areas adjacent to the Naval Academy, taking part in the "Field Practices" subject. 63 In addition, for adults, they have a week of autumn maneuvers in Lugo, another week of 64 winter maneuvers in the Sierra de León at altitudes that exceed a thousand meters in 65 height and, finally, two weeks in maneuvering fields that vary from the two previously 66 mentioned even others such as the Sierra del Retín in Cádiz. It is precisely in Cádiz and 67 other southern cities such as Cartagena, where these students will also be assigned in the 68 future when they graduate as officers of the Military Naval School and where the insola- 69 tion increases considerably compared to Galicia. As can be deduced from all these factors, 70 the Marine Corps student at the Naval Military School is not only exposed during their 71 training period to the impact of ultraviolet radiation (UV) but will also continue to be so 72 throughout their lives. On the other hand, the General Corps and Intendancy students are 73 not left behind on this issue either. In addition to having the same maneuvering periods 74 as their fellow Marine Corps, in addition to the UV radiation that can reach them directly 75 from the Sun, the reflected in sea water must be added, this being a multiplying factor of 76 the effects. detrimental to it, as will be detailed later. that are required of officers of the 77 Navy. Most of these areas of the world are close to the Equator, where the impact of UV 78 radiation is much more dominant, such as regions of Africa or the Middle East."                            One suggestion is to mention that students undergo rigorous training programme which exposes them not only to summer heat and sun but also to winter harsh conditions at high altitudes where the sun is even more intense and UV radiation more concentrated. In addition, marine army corps are subjected to high levels of reflected solar radiation from the sea.

Page 2 Line 90: "uniformity" corresponding to sailors also serves to protect 91 themselves from the Sun, since headwear should be worn outdoors, such as caps in most 92 of the countries. cases, and long-sleeved fire-retardant "work".  Please consider the word "uniform gear" instead of "uniformity"  and "material" instead of "work".

Page 2 Line 94: "epidemic" is used for infectuous diseases. This is not the case. Please replace the word with "high".

Page 3 Line 126: "strip" to be replaced by "age band"

Page 3 Line 130: Why Ecuador? Please replace by a more generic term e.g. "a forest"

Page 3 Line 138: "In its most simplified form, SEM can be understood as the combination of path anal-ysis with factor analysis [19,20]. The inference that the SEM enables the understanding of complex patterns of interrelationships.   Please reconsider e.g.: "In its most simplified form, SEM can be understood as the combination of path analysis with factor analysis [19,20], to enable the understanding of complex patterns of interrelationships."

Page 3 Line 145: "causes" to be replaced by "factors".

Page 4 Lines 173-176: There are too many sensitive information details: "The students of the Spanish Marine Corps spent a week of maneuvers in altitudes 173 that exceed 1000 meters above sea level. Moreover, in summer, during the months of June 174 and July, they also spent two weeks of maneuvers in different locations that can vary from 175 the Army Maneuvering and Shooting Range in Parga, Lugo, to the Sierra del Retín, in 176 Cádiz, where the impact of radiation is much greater."  The gist is: "This study has focused on a select of students from a military school in Spain that was undergoing outdoor training at the time of this study".

Table 1: The font of the third column is not the same as the rest of the text. e.g. "1. If a military member does not protect himself or herself from the sun, the risk of suffering from skin cancer in the future is: very low (1) to very high (5)". Please change the font type.

Page 6 Line 191: "To obtain these results, the questions have been carried out so that it is the respondent 191 who decides to give the parameters this importance and influence. It is worth mentioning 192 that to answer the survey, at first, the students reported the lack of examples to under- 193 stand the formulation of the questions, so that, after a first unsuccessful attempt, an im- 194 provement was made in the way of evaluating the questions. relationships, without 195 changing either the question format or the scale to be used, this being from 0 to 3. How- 196 ever, all the parameters to be related were grouped together in the same question, while 197 in the first survey, each one was formulated independently. In addition, examples of pos- 198 sible answers were written next to each question, thus facilitating the respondent's re- 199 sponse."  Please consider re-phrasing: "The first version of the questionnaire was found to be difficult to comprehend and therefore a number of improvements were made to it such as better consolidation of the questions and provision of some examples to better explain the questionnaire."

Page 6 Line 203: "hits" should be THIS

Page 6 line 204: This sentence is quite confusing and not necessary: "The thesis is not clear enough about these rela- 204 tionships. It just mentions a reference and a few adaptations. It just mentions a reference 205 and a few adaptations. It has no means to explain their meaning and dependences (the 206 hypothesis shown in the Figure 1). The questionnaire must be translated to English before 207 explaining the observed variables." Consider deletion. Then try to re-organise the paragraph following it. For example, you can say: "In Figure 1, a high level flow chart of the methodology is provided..... and try to explain Figure 1 in more details.

Page 7 Line 226: Have you explained what these abbreviations are in the text? I don't think so. "ATT, PLN and SOC".  One good example that you have made is as follows: "exogenous variables Susceptibility (SUS) severity (SEV)" in line 237. Please use a similar approach for explaining ATT, PLN and SOC.

Page 7 Line 231: Leave a space before "and". ........r [27]and, t......

Page 7 Line 240: This sentence is not clear "What confirms that if protecting from the sun avoids the risk of future skin cancer and in 240 the event of a positive diagnosis of this type of cancer, this would be a serious situation, 241 wearing clothing with sun protection would be an effective way for someone to protect 242 from the sun.".

The printing quality of Figure 2 is not sufficiently good. Try to make a clearer diagram. For example avoid copying and pasting from a pdf file.

Page 8 Line 263: "on the other hand" to be replaced by "while".

Table 2 text font size is too large compared to the rest of the paper.

Page 9 Line 290: Please replace "he" with "one" to be gender neutral.

Author Response

The modifications are highlighted in the manuscript: yellow (answers reviewer 1) and blue (answers reviewer2).

QUESTION 1
==========

The abbreviation IUV should probably be UVI = Ultraviolet Index.

Page 2 line 38: "Without going any further, in countries very exposed to the Sun such as Mali, where the Marine Corps currently has deployed troops, or Iraq (Spanish Ministry of Defense, 2020), where there are also troops, UV radiation can be very dangerous, reaching very high IUV levels." Kindly reconsider this sentence: First, there is no need to pin point places where troops are deployed. One can just say in places where the solar intensity is high, and especially within the sun belt region (between Latitude 35° S and 35° N), troops may be subjected to high and prolonged exposure to solar UV intensity.

REPLY

A modification of the paragraph has been done.

ACTION

... with this problem. Their maneuvers can be performed in places where the solar intensity is high like the sun belt region between 35º S and 35 º N where troops may be subjected to high ultraviolet index levels...

QUESTION 2
==========

There is a lot of unnecessary detail, which may be subject to confidentiality. Did you get permission from the academy to publish such details? One week training, full day in the sun, etc....? I suggest that you review the whole paragraph: "In most units of the Spanish Navy, the activities carried out there are largely abroad, 58 especially during periods of instruction and training [4,5]. Even though Galicia is one of the regions of Spain that receives the least sunstroke per year, this work has aimed to focus on the students of the Marín Naval Military School (ENM). In the case of Marine Infantry students, they spend at least one full day in the field each week of the academic year, in different areas adjacent to the Naval Academy, taking part in the "Field Practices" subject. In addition, for adults, they have a week of autumn maneuvers in Lugo, another week of winter maneuvers in the Sierra de León at altitudes that exceed a thousand meters in height and, finally, two weeks in maneuvering fields that vary from the two previously mentioned even others such as the Sierra del Retín in Cádiz. It is precisely in Cádiz and other southern cities such as Cartagena, where these students will also be assigned in the future when they graduate as officers of the Military Naval School and where the insolation increases considerably compared to Galicia. As can be deduced from all these factors, the Marine Corps student at the Naval Military School is not only exposed during their training period to the impact of ultraviolet radiation (UV) but will also continue to be so throughout their lives. On the other hand, the General Corps and Intendancy students are not left behind on this issue either. In addition to having the same maneuvering periods as their fellow Marine Corps, in addition to the UV radiation that can reach them directly from the Sun, the reflected in sea water must be added, this being a multiplying factor of the effects. detrimental to it, as will be detailed later. that are required of officers of the Navy. Most of these areas of the world are close to the Equator, where the impact of UV radiation is much more dominant, such as regions of Africa or the Middle East."
One suggestion is to mention that students undergo rigorous training programme which exposes them not only to summer heat and sun but also to winter harsh conditions at high altitudes where the sun is even more intense and UV radiation more concentrated. In addition, marine army corps are subjected to high levels of reflected solar radiation from the sea.

REPLY

The intention of the authors was to publicize the activities of the students and to enhance their value. However, as the reviewer has correctly pointed out, it could be a problem from the confidentiality point of view. Therefore, the suggested correction has been done.

ACTION

...instruction and training [4,5]. Students undergo rigorous training programme which exposes them not only to summer heat and sun but also to winter harsh conditions at high altitudes where the sun is even more intense and UV radiation more concentrated. In addition, marine army corps are subjected to high levels of reflected solar radiation from the sea. Also, their activities can be performed in areas of the world close to the Equator...

QUESTION 3
==========

Page 2 Line 90: "uniformity" corresponding to sailors also serves to protect 91 themselves from the Sun, since headwear should be worn outdoors, such as caps in most 92 of the countries. cases, and long-sleeved fire-retardant "work".  Please consider the word "uniform gear" instead of "uniformity"  and "material" instead of "work".

REPLY

Both replacements has been done.

ACTION

...Despite this, like the military in land operations, the uniform gear corresponding to sailors also serves to protect themselves from the Sun, since headwear should be worn outdoors, such as caps in most of the countries. cases, and long-sleeved fire-retardant material

QUESTION 4
==========

Page 2 Line 94: "epidemic" is used for infectuous diseases. This is not the case. Please replace the word with "high".

REPLY

The correction has been done.

ACTION

...The incidence of skin cancer is continuously increasing and has reached high proportions worldwide...

QUESTION 5
==========

Page 3 Line 126: "strip" to be replaced by "age band"

REPLY

The replacement has been done.

ACTION

...present in civilian citizens, especially the age band between 55 and 59 years of age

QUESTION 6
==========

Page 3 Line 130: Why Ecuador? Please replace by a more generic term e.g. "a forest"

REPLY

The replacement has been done.

ACTION

...since the simple fact of being assigned to an area closer to a forest, in the sea, the desert or higher...

QUESTION 7
==========

Page 3 Line 138: "In its most simplified form, SEM can be understood as the combination of path anal-ysis with factor analysis [19,20]. The inference that the SEM enables the understanding of complex patterns of interrelationships.   Please reconsider e.g.: "In its most simplified form, SEM can be understood as the combination of path analysis with factor analysis [19,20], to enable the understanding of complex patterns of interrelationships."

REPLY

The suggested correction has been done. The new reference numbers are produced by the added bibliography.

ACTION

In its most simplified form, SEM can be understood as the combination of path analysis with factor analysis [28,29] to enable the understanding of complex patterns of interrelationships. The inference…

QUESTION 8
==========

Page 3 Line 145: "causes" to be replaced by "factors".

REPLY

The replacement has been done.

ACTION

...which are supposedly the factors that cause a set of observed variables (measured or indicators) to vary, are usually...

QUESTION 9
==========

Page 4 Lines 173-176: There are too many sensitive information details: "The students of the Spanish Marine Corps spent a week of maneuvers in altitudes 173 that exceed 1000 meters above sea level. Moreover, in summer, during the months of June 174 and July, they also spent two weeks of maneuvers in different locations that can vary from 175 the Army Maneuvering and Shooting Range in Parga, Lugo, to the Sierra del Retín, in 176 Cádiz, where the impact of radiation is much greater."  The gist is: "This study has focused on a select of students from a military school in Spain that was undergoing outdoor training at the time of this study".

REPLY

The rewritten of the sentence has been done.

ACTION

This study has focused on a select of students from a military school in Spain that was undergoing outdoor training at the time it was carried out. These respondents...

QUESTION 10
==========

Table 1: The font of the third column is not the same as the rest of the text. e.g. "1. If a military member does not protect himself or herself from the sun, the risk of suffering from skin cancer in the future is: very low (1) to very high (5)". Please change the font type.

REPLY

The format issued has been corrected.

ACTION

Please, consult Table 1.

QUESTION 11
==========

Page 6 Line 191: "To obtain these results, the questions have been carried out so that it is the respondent who decides to give the parameters this importance and influence. It is worth mentioning that to answer the survey, at first, the students reported the lack of examples to understand the formulation of the questions, so that, after a first unsuccessful attempt, an improvement was made in the way of evaluating the questions. relationships, without changing either the question format or the scale to be used, this being from 0 to 3. However, all the parameters to be related were grouped together in the same question, while in the first survey, each one was formulated independently. In addition, examples of possible answers were written next to each question, thus facilitating the respondent's response."  Please consider re-phrasing: "The first version of the questionnaire was found to be difficult to comprehend and therefore a number of improvements were made to it such as better consolidation of the questions and provision of some examples to better explain the questionnaire."

REPLY

The suggested re-phrasing has been done and added to the previous paragraph.

ACTION

...military and subordinates. The first version of the questionnaire was found to be difficult to comprehend and therefore a number of improvements were made to it such as better consolidation of the questions and provision of some examples to better explain the questionnaire.

QUESTION 12
==========

Page 6 Line 203: "hits" should be THIS

REPLY

The typo has been corrected.

ACTION

...At this point, an explanation about the latent variables, unobserved exogenous...

QUESTION 13
==========

Page 6 line 204: This sentence is quite confusing and not necessary: "The thesis is not clear enough about these relationships. It just mentions a reference and a few adaptations. It just mentions a reference and a few adaptations. It has no means to explain their meaning and dependences (the hypothesis shown in the Figure 1). The questionnaire must be translated to English before explaining the observed variables." Consider deletion. Then try to re-organise the paragraph following it. For example, you can say: "In Figure 1, a high level flow chart of the methodology is provided… and try to explain Figure 1 in more details.

REPLY

The sentence has been modified according to the reviewer's suggestions.

ACTION

... observed variables must be done. In Figure 1, a high level flow chart of the methodology is provided. Three latent and five observed variables were applied in this model. Attitude, Social Influence and Action Planning were the latent variables. Perceived Susceptibility, Perceived Severity, Self-Efficacy, Intention and Sun Protection Clothing Use the observed variables.. This workflow model is based on the work of Van Osch et al. [22]. As indicated above, the questionnaire and the model of the relationships between input variables used in the research by Van Osch et al. [22] had to be translated into Spanish and adapted to the objective of the study: to analyse the effectiveness of the use of UV protective clothing in students linked to the military. In other words, Figure 1 shows the hypothesis proposed in the model of dependencies between the variables used in this study.

QUESTION 14
==========

Page 7 Line 226: Have you explained what these abbreviations are in the text? I don't think so. "ATT, PLN and SOC".  One good example that you have made is as follows: "exogenous variables Susceptibility (SUS) severity (SEV)" in line 237. Please use a similar approach for explaining ATT, PLN and SOC.

REPLY

The correspondence of ATT, PLN, SOC abbreviations with their respective names was done in the same sentence used for SUS and SEV.

ACTION

...The exogenous observed variables Susceptibility (SUS) and Severity (SEV), and the latent variables Attitude (ATT), Planning (PLN) and Social Influence (SOC) revealed the absence of multicollinearity...

QUESTION 15
==========

Page 7 Line 231: Leave a space before "and". ........r [27]and, t......

REPLY

The typo has been corrected.

ACTION

...Fornell and Larcker [31] and, therefore, meeting the reliability…

QUESTION 16
===========

Page 7 Line 240: This sentence is not clear "What confirms that if protecting from the sun avoids the risk of future skin cancer and in the event of a positive diagnosis of this type of cancer, this would be a serious situation, wearing clothing with sun protection would be an effective way for someone to protect from the sun."

REPLY

The sentece was modified.

ACTION

This confirms hypotheses H0 and H1 at 10% of significance level, that protecting oneself from the sun prevents the risk of developing skin cancer in the future. Furthermore, in the event of a positive diagnosis of skin cancer, wearing sun-protective clothing would be an effective way of protecting oneself from the sun.

QUESTION 17
===========

The printing quality of Figure 2 is not sufficiently good. Try to make a clearer diagram. For example avoid copying and pasting from a pdf file.

REPLY

Figure 2 was replaced by a SEM diagram adapted by the authors.

ACTION

Please, see the new Figure 2

QUESTION 18
===========

Page 8 Line 263: "on the other hand" to be replaced by "while".

REPLY

The correction has been done. Also, a new section called Discussion was created. The text has been placed in this section.

ACTION

...close to zero, while SOC-INT was not significant...

QUESTION 19
===========

Table 2 text font size is too large compared to the rest of the paper.

REPLY

Effectively, the table font size is bigger than the font of the text.

ACTION

The font size has been downsized from 11 to 9.

QUESTION 20
===========

Page 9 Line 290: Please replace "he" with "one" to be gender neutral.

REPLY

The correction has been done. The paragraph has been placed in Discussion section.

ACTION

...This means that when a person plans to wear an outfit with sun protection, one is likely to convert this idealization into a habit of use...

Reviewer 2 Report

The manuscript "  An assessment on the efficiency of clothing with UV protection 2 among the Spanish Navy School students "is well written and it details the harmful effects that ultraviolet (UV) rays have on the skin of people.

However, kindly find the below comments and provide responses to each of them.

1. Abstract should include some numerical results and significant outcomes.

2. Major Contributions of the work should be highlighted in the last paragraph of the introduction.

3. Maintain the MDPI format to cite the references. Few refs are written in MLA and APA format. please check. for example... (Noar, et al., 2015)

4. Fig.2 must be replaced with a better one.

5. A critical discussion of the significant outcome and the overall study should be reported in a different section.

 6. Provide a comparison with respect to samples and methods to literature.

7. Discuss the current standards in clothing UV protection. 

Author Response

The modifications are highlighted in the manuscript: yellow (answers to reviewer 1) and blue (answers to reviewer2).

QUESTION 1
==========

Abstract should include some numerical results and significant outcomes.

REPLY

New text has been written.

ACTION

... during tactical maneuvers. These relationships were significant with p-values close to zero. However, exogenous variables related to Perceived Susceptibility and Perceived Severity in exposure to sunlight did not represent a significant inf luence when mediated by Self-Efficcacy in use. The results revealed the consequence of awareness about the importance of protecting oneself and the influence that usage habits can have on the military with respect to the decision to purchase uniforms with UV protection.

QUESTION 2
==========

Major Contributions of the work should be highlighted in the last paragraph of the introduction.

REPLY

The last paragraph of the Introduction shows the objective and the major contribution of the present research.

ACTION

The objective of this research is to analyze the behavioral variables associated to the effects on the skin caused by UV rays, denoted by the combined effects of exogenous variables, on their decision to purchase and wear uniforms with UV protection. The students' perception of the attitude of uniforms with UV protection should inspire manufacturers to make this option available, showing that the company's efforts will be rewarded with the purchase decision made by students.

QUESTION 3
==========

Maintain the MDPI format to cite the references. Few refs are written in MLA and APA format. please check. for example... (Noar, et al., 2015)

REPLY

The references have been checked and corrected.

ACTION

...should not be underestimated [26].

QUESTION 4
==========

Fig.2 must be replaced with a better one.

REPLY

Figure 2 was replaced by a SEM diagram adapted by the authors.

ACTION

Please, see the new Figure 2.

QUESTION 5
==========

A critical discussion of the significant outcome and the overall study should be reported in a different section.

REPLY

The critical discussion of the significant outcome was transferred to the new Discussion section.

ACTION

4. Discussion
Regarding the causal relationship the [...] require cognitive reinforcement.

QUESTION 6 
==========

Provide a comparison with respect to samples and methods to literature.

REPLY

A paragraph has been inserted at the end of the Introduction section. Five references have been added.

ACTION

The literature on the use of SEM to analyze problems with UV radiation in people regularly exposed to the external environment has addressed different objectives. A study carried out with 148 male external workers with the objective of confirming that these professionals highly exposed to UV radiation could count on an efficient preventive diagnosis, obtained through interventions with UV photographs [22]. The investigation into the beliefs of 334 students at an American university about UV exposure and sunscreen use, as well as the links with skin cancer and protective measures in a cloudy sky [23]. Adoption of the tripartite theory of body image, reasoned action, health belief model and motivational protection theory to test various behavioral models. In this model, the authors confirmed that intention is a mediator of sociocultural influence and tanning behavior [24]. A model to measure the behavior of 273 parents in protecting their children from the carcinogenic effects of sun exposure. The variables used were intention, attitude, subjective norm, action plan, among others [25]. The use of Latent Class Analysis (LCA) to assess classified homogeneous groups of 90 respondents according to constitutional risk, tanning intention, and sun protection behavior (shade, clothing, and sunscreen) [26]. The collection of a sample of 261 respondents to propose a model based on knowledge, motivation, health literacy and nutritional literacy to test whether knowledge about health and nutrition were associated with attitude towards sun exposure [27].

QUESTION 7
==========

Discuss the current standards in clothing UV protection.

REPLY

Information on the different standards currently in existence was added.

ACTION

...This standard divides UPFs into three groups: those with ratings of 15 to 24, those with ratings of 25 to 39, and those with ratings of 40 and higher, which offer outstanding protection [19]. In addition to the Australian and New Zealand committees, the International Organization for Standardization (ISO) and other global organizations, such as the Commission on Illumination (CIE), have been working on de-veloping UV-protective fabric standard papers. A new standard concerning the specifi-cations for test procedures and labeling of sun-protective clothing (EN 13758) has been created by the European Committee for Standardization (CEN) [19]. The amount of ul-traviolet light that is transmitted or blocked by textile materials designed to be used for UV protection can be also measured using the American Standard (AATCC TM 183, the GBT18830-2009 national standard of the people’s republic of China, and the UV Standard 801 from the International Testing Association in Europe [20,21]. The UPF results by following these measurement standards do not differ significantly [20].

Round 2

Reviewer 2 Report

The authors have provided an adequate explanation of the point raised. I don't have further queries.